# Dyslexics' faster decay of implicit memory for sounds and words is manifested in their shorter neural adaptation

**Sagi Jaffe-Dax[1,2]\*, Or Frenkel[3], Merav Ahissar[1,3]\***

[1]Edmond and Lily Safra Center for Brain Sciences, The Hebrew University of Jerusalem, Jerusalem, Israel; [2]Department of Psychology, Princeton University, Princeton, United States; [3]Psychology Department, The Hebrew University of Jerusalem, Jerusalem, Israel

**Abstract** Dyslexia is a prevalent reading disability whose underlying mechanisms are still disputed. We studied the neural mechanisms underlying dyslexia using a simple frequency-discrimination task. Though participants were asked to compare the two tones in each trial, implicit memory of previous trials affected their responses. We hypothesized that implicit memory decays faster among dyslexics. We tested this by increasing the temporal intervals between consecutive trials, and by measuring the behavioral impact and ERP responses from the auditory cortex. Dyslexics showed a faster decay of implicit memory effects on both measures, with similar time constants. Finally, faster decay of implicit memory also characterized the impact of sound regularities in benefitting dyslexics' oral reading rate. Their benefit decreased faster as a function of the time interval from the previous reading of the same non-word. We propose that dyslexics' shorter neural adaptation paradoxically accounts for their longer reading times, since it reduces their temporal window of integration of past stimuli, resulting in noisier and less reliable predictions for both simple and complex stimuli. Less reliable predictions limit their acquisition of reading expertise.

**\*For correspondence:** sagi.jaffe@ mail.huji.ac.il (SJ-D); msmerava@ gmail.com (MA)

**Competing interests:** The authors declare that no competing interests exist.

## Introduction

Dyslexics are diagnosed on the basis of their persistent difficulties in acquiring peer-level reading skills despite adequate education. Their general reasoning skills are within the normal range (or above), but they consistently show difficulties in some language-related skills such as verbal working memory (e.g. *Torgeson and Goldman, 1977*) and phonological manipulations (which typically also load on short-term memory [e.g. *Landerl et al., 1997*]). Dyslexics also often have higher thresholds in simple perceptual discrimination tasks (*Mcanally and Stein, 1996*; *Witton et al., 1998*; *Hämäläinen et al., 2013*), particularly when administered with serial presentations (*Ben-Yehudah and Ahissar, 2004*; discussed in *Ramus and Ahissar, 2012*). In most of these studies, the responses of participants can be more successful by taking into account the frequency statistics of previous stimuli (*Ahissar et al., 2006*; *Oganian and Ahissar, 2012*).

The putative causes of dyslexics' difficulties on simple serial tasks have been examined in a series of studies using 2-tone frequency discrimination. *Ahissar et al. (2006)* measured the impact of sound regularities on dyslexics' performance. They assessed a well-documented observation (*Harris, 1948*) that listeners use a repeated reference tone as an 'anchor', and that their performance was improved in protocols that included this reference tone compared with a no-reference protocol. The benefit that dyslexics obtained from this repetition was smaller than that obtained by 'good readers'. A similar deficit was found in dyslexics' benefit from repetition of speech sounds. This led

**eLife digest** The term "dyslexia" comes from the Greek for "difficulty with words". People with dyslexia struggle with reading and spelling: they may mix up letters within words and tend to read and write more slowly than others. However, not every symptom of dyslexia is related to literacy. Affected individuals also differ from good readers on simple perceptual tasks, such as distinguishing between tones of different frequencies.

In a series of trials involving discrimination between pairs of tones, a person's performance on each trial will be influenced by the tones presented on previous trials. Both good readers and individuals with dyslexia automatically form a subconscious memory of the tones they hear, and use this memory to guide their performance on subsequent trials. However, people with dyslexia benefit less from this effect than good readers.

Jaffe-Dax et al. have now identified the mechanism that underlies this phenomenon, revealing new insights into how dyslexia influences brain activity. By varying the interval between successive pairs of tones, the experiments showed that the memory of previous tones decays faster in people with dyslexia than in good readers. A similar effect occurs when the stimuli are nonsense words. Both good and poor readers manage to read nonsense words more quickly on their second attempt. However, people with dyslexia benefit less from the previous exposure when the gap between repetitions is longer than a couple of seconds.

Further studies are needed to determine whether and how the faster decay of memory traces for words is related to impaired reading ability in people with dyslexia. One possibility is that the faster decay of memory traces makes it more difficult to predict future stimuli, which may impair reading. An imaging study is underway to investigate where in the brain this rapid decay of memory traces occurs.

to the hypothesis that dyslexics have a deficit in using sound stimuli as perceptual anchors for the formation of sound predictions (*Ahissar et al., 2006*; *Ahissar, 2007*; *Oganian and Ahissar, 2012*).

*Raviv et al. (2012)* extended the protocol-specific account of benefits from stimulus repetition to generate a computational model which takes the experiment's statistics into account. This model assumes that listeners implicitly infer the mean (frequency) even when it is not presented explicitly. The inferred mean of previous stimuli (prior) is combined with the representation of the current stimulus, and forms an integrated percept (posterior). The resulting percept is contracted (biased) towards that mean (the 'contraction bias'; *Woodrow, 1933*; *Preuschhof et al., 2010*). This bias is advantageous when the observation of the current stimulus is noisy, and hence integration with prior knowledge is likely to improve its accuracy. Indeed, in the general population, 'noisier' listeners weigh the prior more than 'less noisy' listeners when integrating the prior with current representation, resulting in their larger contraction bias. *Jaffe-Dax et al. (2015)* found that dyslexics' bias is smaller than controls' even though they tend to be 'noisier' listeners.

In the two-tone frequency discrimination task, the representation of the first tone is contracted towards the prior more than the second tone, because of the noise added to its representation during the encoding and retention in memory through the inter-stimulus time interval. The contraction of the first tone towards the prior can increase the perceived difference between the two tones in the trial, and hence improve discrimination (*Bias+* trials; e.g., trial $t$ in **Figure 1A**); alternatively, it could decrease the perceived difference between the two tones and disrupt performance (*Bias-*; e.g., trial $t - 1$ in **Figure 1A**). The difference in performance between *Bias+* and *Bias-* trials reflects the magnitude of the contraction bias (i.e. context effect). Dyslexics' underweighting of the prior yields a smaller contraction bias, namely a smaller performance difference between these two types of trials (*Jaffe-Dax et al., 2015*).

Using this measure of contraction bias, we studied the dynamics of controls' and dyslexics' benefits from the statistics of recent sound stimuli in both simple discriminations and in oral reading. We found that dyslexics are similarly affected (biased) by recent stimuli, but less affected by earlier stimuli, as expressed both behaviorally and in the dynamics of the compulsory ERP components (N1 and P2) produced by the auditory cortex. These observations suggest that dyslexics' automatic

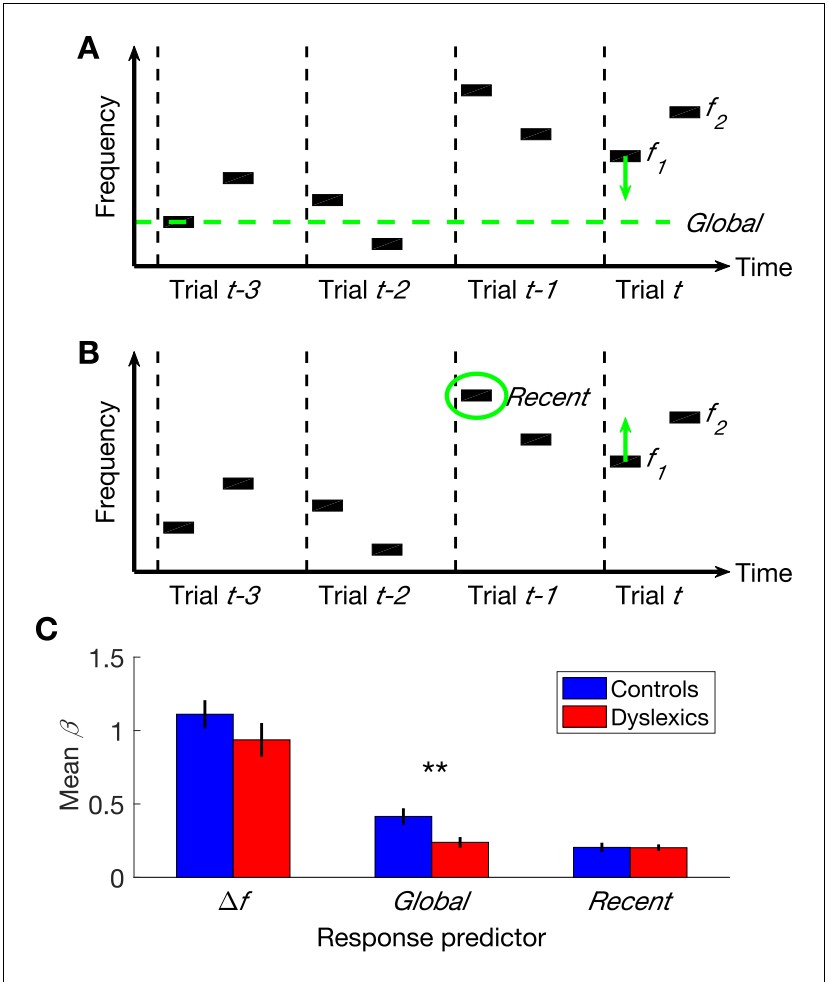

**Figure 1.** The three parameters that additively determine performance in frequency discrimination: *frequency difference within a trial* ($\Delta f$), *Global context* (the difference between the current $f_1$ and the global mean) and *Recent context* (the difference between the current $f_1$ and the previous $f_1$). (**A**) A schematic illustration of the *Global bias:*four trials and the direction of the contraction, which contracts the representation of the first tone in the trial towards the global mean. (**B**) A schematic illustration of the same four trials and the direction of the *Recent bias*, which pulls the representation of this tone towards the recent $f_1$ frequency. (**C**) The estimated contribution ($\beta$) of each of the three parameters to the overall performance of each group. Dyslexics differed from controls only in the smaller magnitude of the contribution of their *Global* predictor (p<0.01). Filled bars denote the mean $\beta$ values; controls in blue and dyslexics in red. Error bars denote SEM.

integration of previous sounds spans shorter time intervals, and is therefore noisier and produces less reliable predictions. We propose that noisier predictions impede dyslexics' acquisition of expert-level performance in a range of tasks, including reading. These observations also pave the way to pursuing the deficits underlying dyslexia in non-human animals.

## Results

### Dissociating the contraction to the most recent trial from the contraction to all previous trials

Performance in the 2-tone frequency discrimination task is affected by two basic components (*Raviv et al., 2012*, *2014*). First, the frequency difference between the two tones composing each trial – participants are more accurate when this difference is larger. Second, the context effects;

namely, the perceptual bias induced by contracting the first tone towards the estimated mean-frequency in previous trials (*Woodrow, 1933*; *Preuschhof et al., 2010*; see Introduction and *Figure 1A–B*). This estimated prior can be viewed as a combination of a recent component and a global component. The most recent factor is the frequency of the first tone of the previous trial (*Figure 1B*), whose impact on the estimated prior is substantially larger than that of any other previous trial ('recency effect'; *Schab and Crowder, 1988*; *Stewart and Brown, 2004*; *Fischer and Whitney, 2014*). The global prior (illustrated in *Figure 1A*) is the mean frequency of all previous trials.

As described above, dyslexics' discrimination behavior is less affected by previous trials (*Jaffe-Dax et al., 2015*). We now aimed to decipher whether dyslexics' deficit encompasses both the recent and the global context effects. To test this, we designed a novel protocol for the two-tone frequency discrimination task, which allowed us to assess the contribution of recent and earlier trials' on performance separately. Unlike in most sequences, where the recent and more global contexts tend to have the same direction, and are therefore very difficult to dissociate, in the new sequence, the recent and the global contexts were not correlated (we ensured that the directions of the local and global effects were not correlated, as described in Materials and methods).

The overall accuracy of the two groups in this task did not differ (controls' mean % correct ± SEM = 75.3 ± 1.6, dyslexics' = 73 ± 1.9; z = 1.3, *n.s.* Mann-Whitney U test), as predicted given that the use of experiment's statistics was not expected to be significantly beneficial in this stimulus series (replicating *Ahissar et al., 2006*). We calculated a GLM model with three predictors ($\beta$s) for each participant (n = 60; 30 control and 30 dyslexic participants) by estimating the magnitude of the contribution of each of the following components to participants' responses: (1) frequency difference in the current trial; (2) contraction bias of the first tone toward the global mean of previous trials; and (3) contraction bias of the first tone towards the first tone of the most recent trial. Dyslexics' frequency difference predictor did not differ from controls' (*Figure 1C*; z = 1.6, *n.s.*). Namely, the frequency difference within the trial had a similar impact on the response in the two groups (similar levels of sensitivity). The impact of the context effect of the most recent trial (ITI ≈ 1.5 s) was also similar in the two groups (z = 0.05, *n.s.*). However, dyslexics' bias towards the global mean was significantly smaller than controls' (z = 2.7, p<0.01). Indeed, the difference in controls between the contributions of the *Global* (all previous trials except the most recent one) and *Recent* contexts was larger than that found in dyslexics (z = 2.4, p<0.05. Mann-Whitney U tests). Importantly, in both groups, the contribution of *Recent* context was significantly above zero (controls: z = 4.4, p<0.0001; dyslexics: z = 4.7, p<0.00001; Wilcoxon tests). Thus, floor effect cannot account for this interaction.

## Comparing the behavioral and neural dynamics of context effects

The observation that dyslexics assign less weight than controls to earlier trial statistics could result from either of the following two mechanisms. The first is that dyslexics experience larger interference effects, and that intervening trials mask each other. The second is that dyslexics' memory trace decays faster in time, even without intervening sounds. To dissociate these alternatives, we manipulated the time interval between consecutive trials. We reasoned that if dyslexics' memory trace decays faster, we might be able to track the neural correlate of this behavioral dynamic.

We administered the two-tone frequency discrimination task with four different inter-trial intervals (ITIs; i.e. the intervals between the second tone of the previous trial and the first tone of the current trial), in four separate blocks. We chose ITIs of 1.5, 3, 6 and 9 s (roughly; see Materials and methods), based on previous reports of cortical adaptation duration (N1 and P2 components; *Hari et al., 1982*; *Lu et al., 1992*; *Sams et al., 1993*).

### Assessing the time constant of the behavioral context effect

We quantified behavioral performance in terms of sensitivity to the difference between the two tones. As a measure of sensitivity, we used $d'$, which was calculated as the difference between the correct rate and the incorrect rate, transformed by inverse cumulative standard distribution function (*Macmillan and Creelman, 2004*): $d' = \Phi^{-1}(HR) - \Phi^{-1}(FA)$. For both groups, performance was better at longer ITIs (mean $d'$ ± SEM for the four ITIs was 0.48 ± 0.05, 0.57 ± 0.05, 0.71 ± 0.06 and 0.78 ± 0.06, respectively; $\chi^2$=54.2, p<$10^{-10}$. Friedman test). The improvement in $d'$ with longer ITIs is probably related to easing the stress invoked by short ITIs.

Across all ITIs, controls' ($n$ = 23) performance was slightly more accurate than dyslexics' performance ($n$ = 25; mean $d'$ ± SEM of controls = 0.71 ± 0.07, dyslexics = 0.56 ± 0.07; $z$ = 2.0, p<0.05. Mann-Whitney U test). There was no significant interaction between group and ITI (group x ITI; $F_{3,46}$ = 1.4, *n.s.* Repeated Measures ANOVA test). Controls' slightly better performance is interesting, particularly as no group difference was found for the same participants in Experiment 1. The group difference was small, and was within the range of the estimated group variability. It may, however, reflect slightly better learning of the task's characteristics by the control participants.

We calculated the behavioral context effect as the difference (in $d'$) between performance on trials for which contraction to the prior was beneficial (*Bias+*) and performance on trials for which this contraction was disruptive (*Bias-*). In this experiment, we did not statistically dissociate between the impact of the most recent trial and that of earlier ones (which we did in Experiment 1). But the actual inter-trial intervals (i.e. the periods of time since the previous trial) were manipulated. In Experiment 1, the Local and Global effects were highly correlated, as in a typical random sequence of trials. We thus relate to them as a unified prior effect. Previous studies have found that the impact of earlier stimuli decayed exponentially as a function of the number of trials which had passed (*Raviv et al., 2012*) or the time interval between trials (*Lu et al., 1992*) or both (*Fischer and Whitney, 2014*). Thus, to quantify the dynamics of this effect, i.e. its magnitude as a function of the time interval from previous trials, we fitted the calculated bias ($\Delta d'$; difference between individual $d'$ in *Bias+* trials and in *Bias-* trials) as a function of the ITI to an exponential decay model: $\Delta d'(t) = \alpha + \beta exp(-t/\tau)$, where $\alpha$ denotes the estimated $\Delta d'$ at $t \to \infty$ (asymptotic level); $\beta$ denotes the difference between the $\Delta d'$ at $t = 0$ and at $t \to \infty$ (decay magnitude); and $\tau$ denotes the time it takes for $\Delta d'$ (at $t = 0$) to decay to $1/e$ (~37%) of its initial value (temporal slope parameter; a small $\tau$ indicates fast decay); $t$ is the ITI.

Evaluating the dynamics of the contraction bias as a function of ITI showed that its decay was significantly faster among dyslexics, as illustrated in *Figure 2B* (dashed line) and *Figure 2C bottom* (controls' $\tau$ = 6 ± 0.9 s, dyslexics' $\tau$ = 2.9 ± 0.8 s; mean ± SEM; $z$ = 2.2, p<0.05). The two other parameters did not differ between the two groups ($\alpha$ for controls = 0.7 ± 0.1 $d'$, $\alpha$ for dyslexics = 0.5 ± 0.1 $d'$; $z$ = 1, *n.s.*; $\beta$ for controls = 2.3 ± 0.8 $d'$, $\beta$ for dyslexics = 4 ± 0.9 $d'$; $z$ = 1.9, *n.s.* Mann-Whitney U tests), in line with our hypothesis.

## Assessing the dynamics of the neural trace

We hypothesized that the neural mechanism that mediates the inference of the prior frequency is neural adaptation, which is an automatic, stimulus-specific form of memory (*Ulanovsky et al., 2003*). Importantly, it has been found that the time constant of context effects in sound discrimination is the same as that of cortical adaptation to sounds (*Lu et al., 1992*). The two prominent ERP components that are automatically produced by the auditory cortex are N1 and P2 (*Sheehan et al., 2005*; *Mayhew et al., 2010*), which peak at 100 and 200 ms from stimulus onset, respectively. Whether they are produced by a single cortical generator (evolving in time) or by two is still debated (*Lanting et al., 2013*). Both have been found to have some sensitivities to stimuli statistics (*Tremblay et al., 2010*; *Herrmann et al., 2015*). However, in our previous study, where we assessed sensitivity to the frequency prior (*Jaffe-Dax et al., 2015*), only the magnitude of P2 showed such sensitivity. We therefore hypothesized that P2 would be directly associated with the contraction bias. Still, as both components were reported to be sensitive to stimuli statistics, we analyzed the dynamics of both components. We hypothesized that for each group, behavioral context effects and cortical adaptation (N1 and/or P2) would have similar time constants and that these time constants would be shorter among dyslexics.

In both groups, the magnitude of the P2 responses was smallest in the block with the shortest ITI (1.5 s) when P2 adaptation was substantial, and increased (recovered) with longer ITIs, as adaptation gradually decayed ($\chi^2$ = 65.7, p<$10^{-13}$. Friedman test). Among dyslexics, P2 reached its peak magnitude by 6 s ITI (*Figure 2A*, red dot-dashed line), whereas controls' P2 was larger at 9 s ITIs than at 6 s ITIs (*Figure 2A*, blue dotted line). To quantify the dynamics of P2 adaptation under active discrimination, we fitted each individual's areas of P2 elicited by the first tone in each of the four ITIs to the same equation of exponential decay (*Figure 2B*). *Figure 2C top* shows the three fitted parameters for each of the two groups. It shows that the groups did not differ in the estimated amplitudes of P2 at either maximal or minimal adaptation ($\alpha$ for controls = 396 ± 47 $\mu V$ x $ms$, $\alpha$ for dyslexics = 322 ± 33 $\mu V$ x $ms$; $z$ = 1.1, *n.s.*; $\beta$ for controls = −696 ± 99 $\mu V$ x $ms$, $\beta$ for dyslexics =

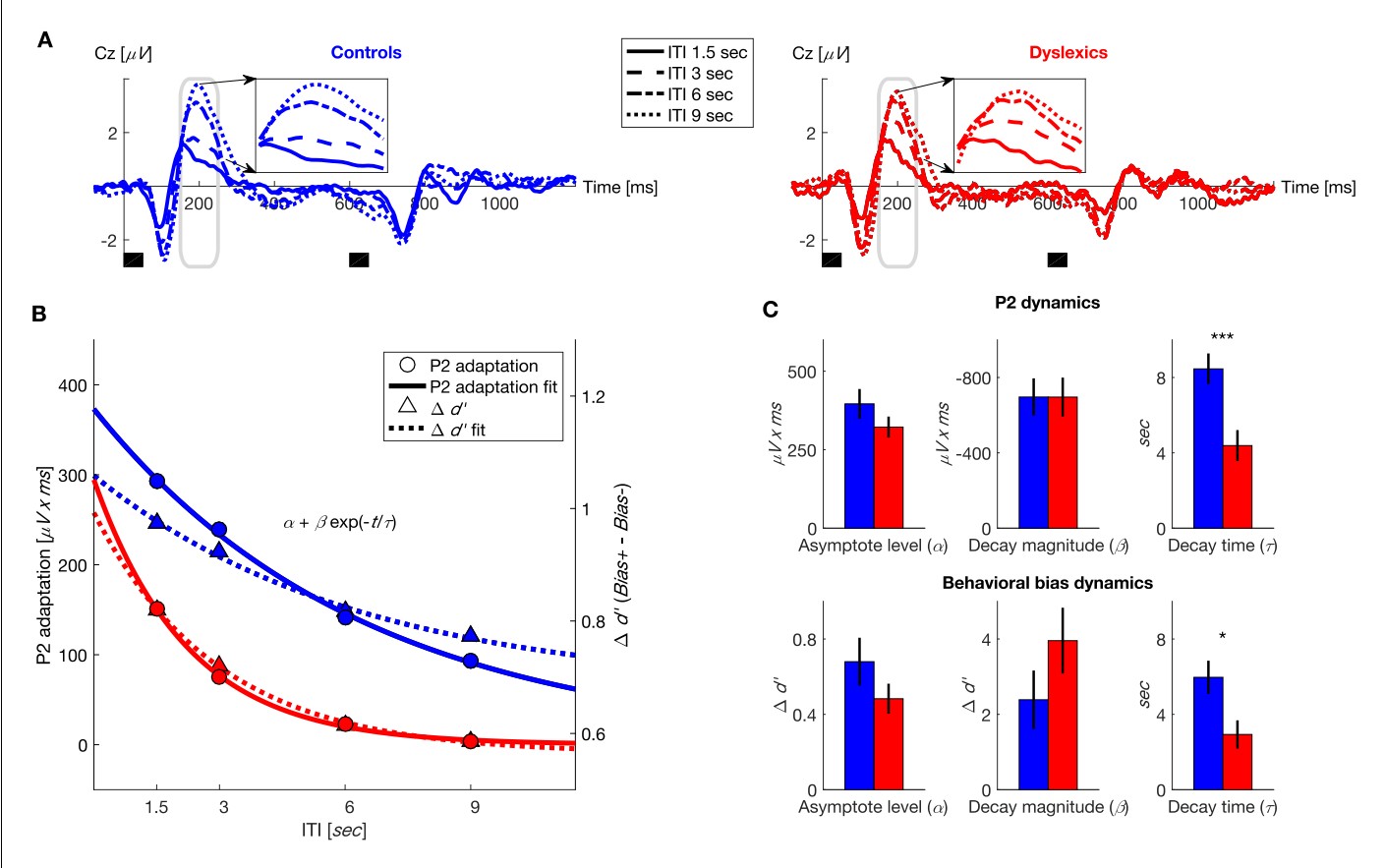

**Figure 2.** Dyslexics' decay of both neural adaptation and behavioral contraction bias was faster than controls'. (**A**) Grand average ERPs, plotted separately for blocks with different ITIs, for controls (n = 23; blue, left) and dyslexics (n = 25; red, right). Timing of the two tones in each trial is denoted by the short black bars under the plots. *Insets*: P2 range (denoted in gray) enlarged. Dyslexics' P2 area was similar for 6 and 9 s ITIs, whereas controls' P2 was larger for the 9 s interval. (**B**) The decrease in P2 adaptation (solid lines, circles, left scale; estimated magnitude at asymptote minus fitted curve) and the decrease in contraction bias (dashed lines, triangles, right scale; reflecting implicit memory decay) as a function of ITI. Symbols denote groups' means, and plotted curves were fitted to these means. (**C**) Groups' means and SEMs of the individually fitted parameters (to an exponential decay) of both P2 adaptation (top) and behavioral contraction bias (***bottom***). In both, dyslexics differed from controls only in the estimated rate of decay (P2: p<0.0005; Δd': p<0.05).

−696 ± 103 $\mu V \times ms$; z = −0.1, *n.s.* Mann-Whitney U tests). However, as with behavior, the rate at which adaptation decayed was significantly faster among dyslexics (τ for controls = 8.5 ± 0.8 s, τ for dyslexics = 4.4 ± 0.8 s; mean ± SEM; z = 3.5, p<0.0005. Mann-Whitney U test).

As was evident when comparing the time constant of the recovery of P2 (time constant of its adaptation) and the time constant of the behavioral context effect, the two were similar, though P2 decay was slightly (but not significantly; z for controls = 0.2, *n.s.*; z for dyslexics = 0.5, *n.s.* Wilcoxon tests) slower than the behavioral time constant. This similarity was also found in the dyslexic group. *Figure 2B* plots the estimated dynamics (the fitted $\alpha + \beta exp(-t/\tau)$ curves) of P2 adaptation and the behavioral contraction bias in the same graph (fitted to groups' means). For clarity, the dynamics of the ERP component are plotted as decaying to zero, i.e. the fitted curve was subtracted from α (the asymptotic level of maximal P2 magnitude) to depict the decay of adaptation. Dyslexics' behavioral and P2 adaptation curves were also similar. Importantly, dyslexics' rate of memory decay, exemplified both behaviorally and in their adaptation dynamics, was faster than controls' (*Figure 2C*).

In addition to examining our *a priori* hypothesis regarding P2, we analyzed the dynamics of N1 adaptation because other studies have found that this ERP component is also sensitive to experiments' stimulus variables (e.g. *Herrmann et al., 2015*). Indeed, dyslexics' shorter adaptation was also found in this ERP component. As illustrated in *Figure 3A and B*, dyslexics' N1 almost fully

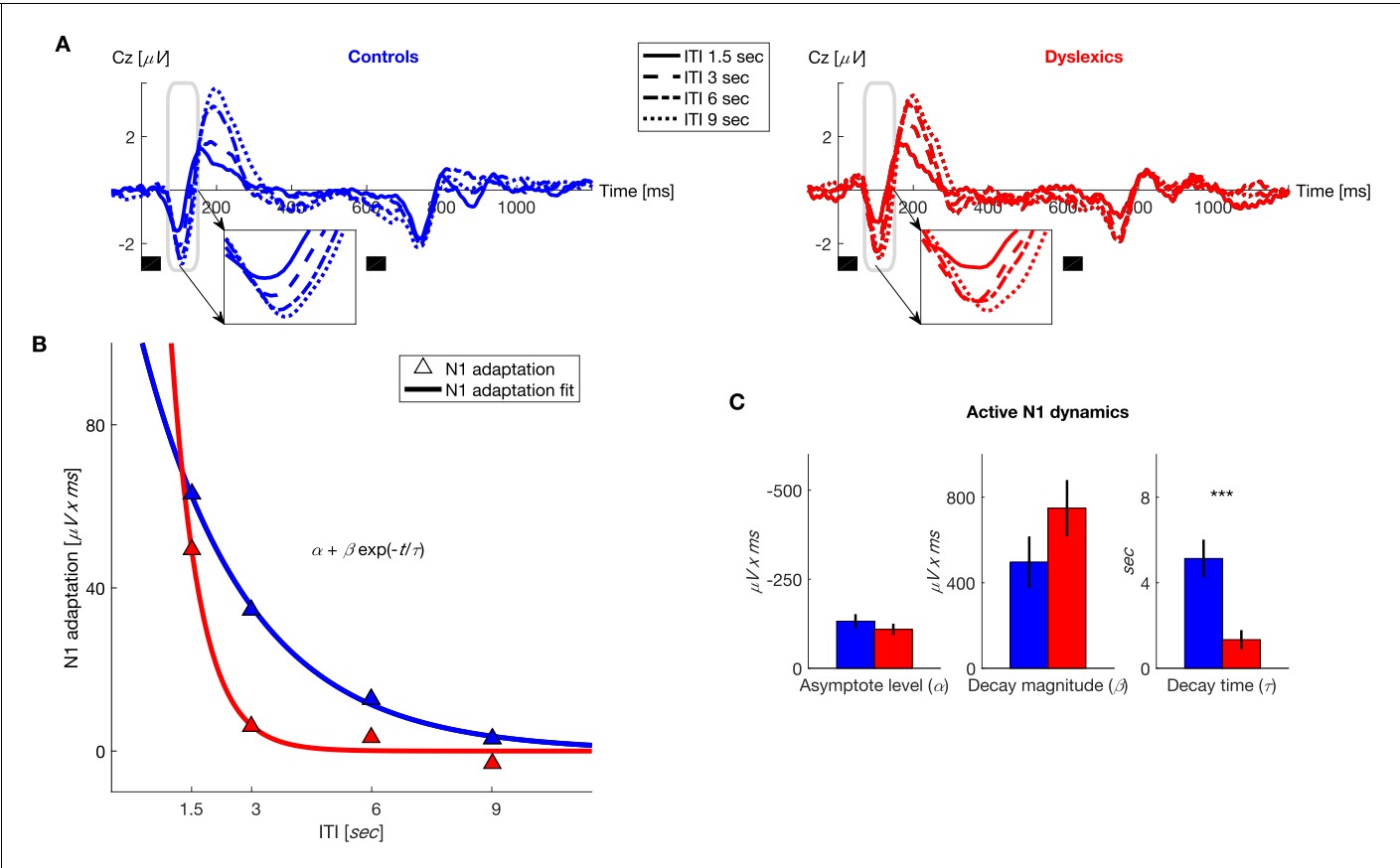

**Figure 3.** Dyslexics' decay of N1 adaptation was faster than that of controls. (**A**) Grand average ERPs, plotted separately for blocks with different ITIs, for controls (n = 23; blue, left) and dyslexics (n = 25; red, right). Timing of the two tones in each trial is denoted by the short black bars under the plots. *Insets*: N1 range enlarged. Dyslexics' N1 almost reached its full magnitude at an ITI of 3 s, whereas controls' N1 reached its near-full magnitude only at an ITI of 6 s. (**B**) The decrease in N1 adaptation (circles) and the fitted exponential decay model (lines) as a function of ITI (curves were fitted to groups' means). (**C**) Groups' mean and SEMs of individually fitted estimated parameters of the exponential decay model of N1 recovery from adaptation. Only $\tau$ significantly differed between the two groups (p<0.0001).

recovered (regained full magnitude) at an ITI of 3 s, whereas the N1 at this ITI was still substantially 'adapted' in controls. Fitting the same exponential model, we found here too that only the rate of decay differed between the two groups (*Figure 3C*; $\tau$ for controls = 5.1 ± 0.8 s, $\tau$ for dyslexics = 1.3 ± 0.4 s; mean ± SEM; z = 4.2, p<0.0001. Mann-Whitney U test), whereas the two other estimated parameters were similar ($\alpha$ for controls = −131 ± 20 $\mu V$ x ms, $\alpha$ for dyslexics = −109 ± 16 $\mu V$ x ms; z = 0.7, n.s.; $\beta$ for controls = 497 ± 120 $\mu V$ x ms, $\beta$ for dyslexics = 750 ± 132 $\mu V$ x ms; z = 1.1, n.s. Mann-Whitney U tests). Across groups, N1 adaptation was significantly shorter than that of P2 (z = 3.7, p<0.0005. Wilcoxon test). There was no interaction between group and ERP component (N1/P2) rate of decay of adaptation (z = 0.7, n.s. Mann-Whitney U test). Finally, given that the dynamics ($\tau$) of the behavioral contraction bias was in between that of P2 and N1 adaptation dynamics, but did not significantly differ from either, the question whether it is P2 or N1 (or a combined process) which best captures the neural processes underlying the behavioral contraction bias remains open.

## Comparing the dynamics of passive adaptation between controls and dyslexics

After finding that adaptation under active performance decays faster among dyslexics, we asked whether this group difference was an automatic characteristic that could also be detected in the absence of a behavioral task. To evaluate the dynamics of dyslexics' passive adaptation, we

administered a sequence of pure tones in four blocks, each with a different ISI, while participants watched a silent movie (ISIs: 2, 3.5, 6.5 and 9.5 s; same as the temporal interval between trial's onsets in the active condition).

To quantify the dynamics of adaptation of P2 and N1, we fitted the P2 and N1 areas to the exponential decay model described above. We found that controls' P2 recovery was significantly slower than dyslexics'. Importantly, the only significant difference between the two groups was in τ (τ for controls = 5.4 ± 0.8 s, τ for dyslexics = 3.3 ± 0.7 s; mean ± SEM; z = 3.3, p<0.005. Mann-Whitney U test; *Figure 4C*). The other two parameters did not differ between the two groups (α for controls = 490 ± 73 μV x ms, α for dyslexics = 410 ± 44 μV x ms; z = −0.04, n.s.; β for controls = −750 ± 112 μV x ms, β for dyslexics = −724 ± 100 μV x ms; z = −0.1, n.s. Mann-Whitney U tests).

N1 adaptation was only marginally shorter among dyslexics (τ for controls = 7.2 ± 0.9 s, τ for dyslexics = 4.1 ± 0.9 s; z = 1.9, p=0.06. α for controls = −132 ± 20 μV x ms, α for dyslexics = −114 ± 16 μV x ms; z = 0.37, n.s. β for controls = 497 ± 120 μV x ms, β for dyslexics = 615 ± 132 μV x ms; z = 0.02, n.s. mean ± SEM; Mann-Whitney U tests). In line with previous studies of passive presentation (*Hari et al., 1982*), we did not find a difference between the time scales of adaptation of P2 and N1 (z = 1.9, n.s. Wilcoxon test) nor was there an interaction of group X ERP component (z = 1.4, n.s. Mann-Whitney U test).

*Figure 4A* shows the average ERP responses in the different blocks, for controls (n = 23; blue, left) and dyslexics (n = 25; red, right), respectively. It shows P2, as only P2 showed a significant group difference (insets in *Figure 4A*). In both groups, P2 responses were the smallest in the block

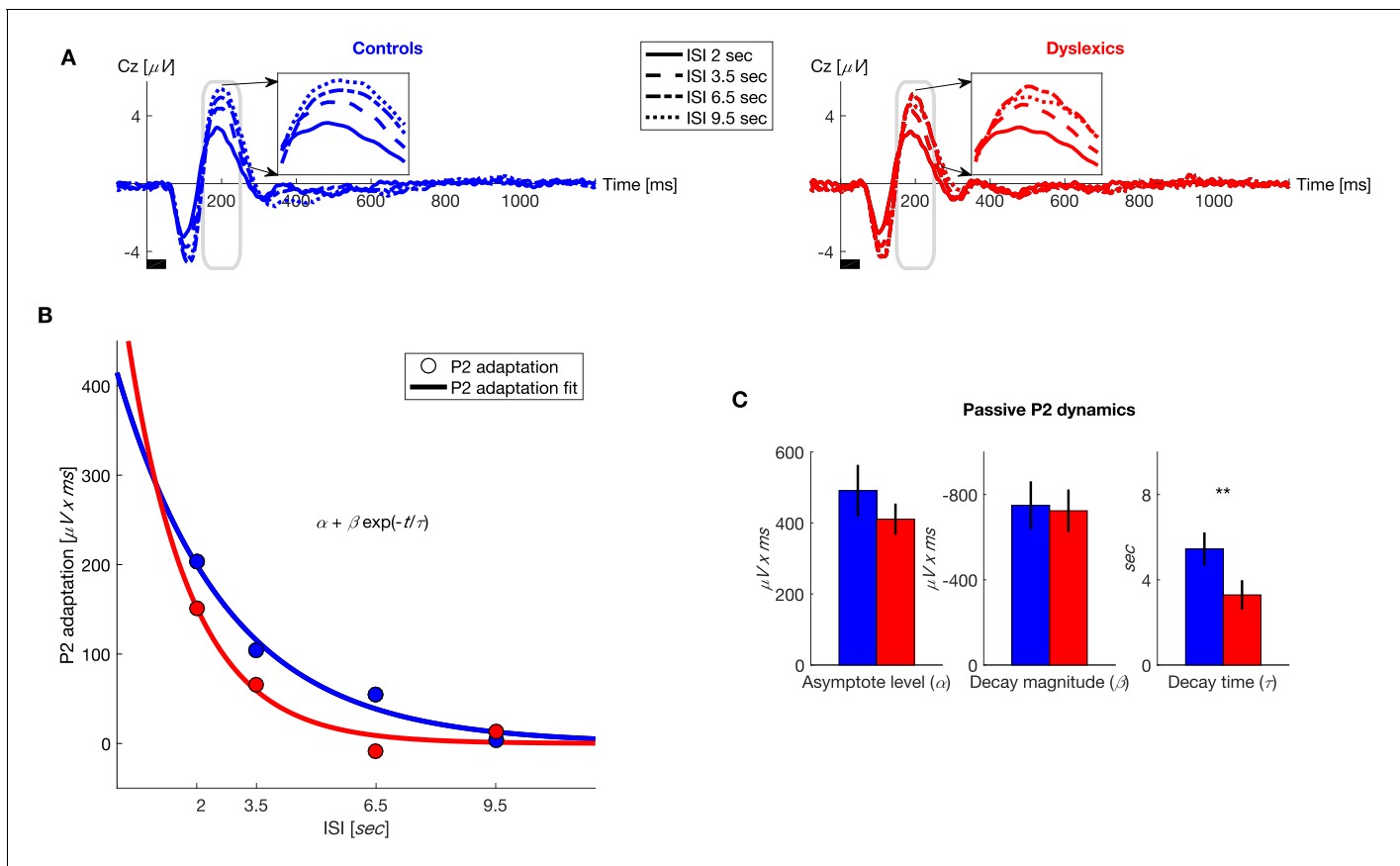

**Figure 4.** Under passive conditions, dyslexics' P2 adaptation is shorter than controls'. (A) Grand average ERPs, plotted separately for blocks with different ISIs, for controls (blue, left) and dyslexics (red, right). *Insets*: P2 range enlarged. Dyslexics' P2 area was largest for the 6.5 s ISI, whereas controls' P2 was largest for the 9.5 s ISI. (B) The magnitude of P2 adaptation as a function of time from the previous stimulus (ISI) under passive condition (circles) fitted with an exponential decay model (solid line). Controls in blue; dyslexics in red. (C) Groups' mean and SEM of individually fitted estimated parameters of P2 recovery from adaptation. Dyslexics differed from controls only in the estimated rate of decay (p<0.005).

with the shortest ISI (2 s), when P2 adaptation was still substantial, and increased (recovered) with longer ISIs, as adaptation gradually decayed ($\chi^2$ = 60.3, p<10$^{-12}$. Friedman test). Among dyslexics, P2 reached its peak magnitude by 6.5 s ISI (*Figure 4A*, red dot-dashed line), whereas controls' P2 was larger at 9.5 s than at 6.5 s ISIs (*Figure 4A*, blue dotted line).

Interestingly, among controls, the estimated duration of P2 adaptation was longer in the active than in the passive condition (P2: $z$ = 3, p<0.005). This was not observed in the dyslexic group (P2: $z$ = 0.5, *n.s.* Wilcoxon tests). The interaction (in the estimated time constant of adaptation) between group and condition (passive/active) was only marginally significant ($z$ = 1.8, p=0.07 Mann-Whitney U tests). Thus, controls' but not dyslexics' P2 adaptation was prolonged by introducing a behavioral context. We did not find such a prolongation by behavioral context on the time scale of N1 adaptation.

Overall, we found a consistent group difference under both passive and active conditions in the dynamics of P2 adaptation, in line with our *a priori* hypothesis (based on *Jaffe-Dax et al., 2015*) regarding P2. We also found a significant group difference in the dynamics of N1 when listeners were actively performing the task. But, a significant group difference under passive conditions was found only for P2.

## Comparing the dynamics of context effects in reading rate

As shown above, dyslexics benefit less from recent sound statistics. We hypothesized that a similar observation would be found for recent encounters with complex sound stimuli and particularly with novel syllabic combinations. If demonstrated, this correlate would provide support for the relevance of the observations of Experiments 1–2 to the context of reading. To assess the dynamics of the simplest context effect in reading, we administered an oral reading task in which participants were asked to read single simple disyllabic non-words aloud, and we measured repetition effects. The non-words were presented on a screen, one at a time, and participants were asked to read them as quickly (and accurately) as they could. Consecutive non-words were presented at an individually paced rate. Non-words were repeated throughout the block at various intervals (i.e. with a different number of other intervening non-words), as schematically illustrated in *Figure 5A*.

Participants' response times (RTs, measured as vocal response onset) were shorter in the second presentation of the same non-word. However, the magnitude of this improvement declined as the time interval from the previous presentation of the same non-word increased (i.e. when there were

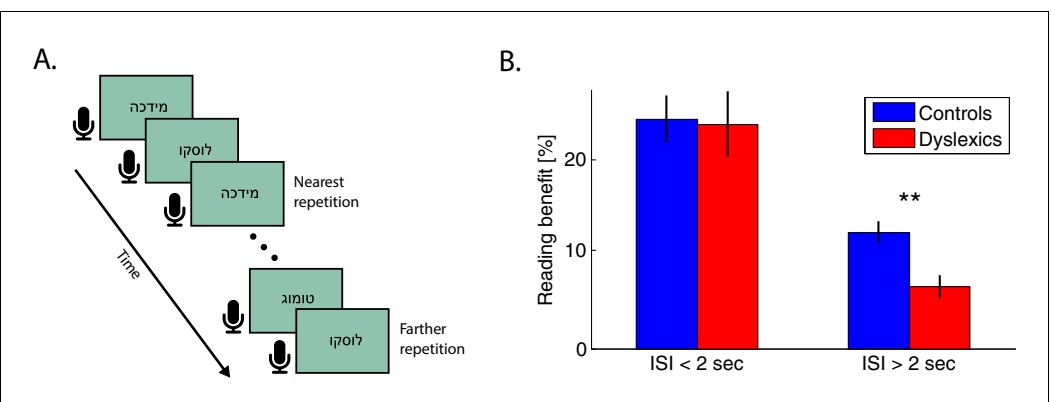

**Figure 5.** Dyslexics' benefit from a previous exposure to the same non-word decayed faster than controls'. (**A**) Schematic illustration of the reading task. Subjects were asked to read the non-words aloud as quickly as possible. Presentation switched to the next word with the subject's voice offset. The closest repetition of the same non-word was with one intervening non-word (i.e. an ITI of <2 s). (**B**) Benefit in RTs (response times from visual word presentation to vocal onset) as a function of the time interval between the first and second presentation of the same non-word. Improvement was calculated as the difference in RT between the first and second presentation of the same non-word in the block. At very short intervals (<2 s), the benefit was similar for both groups. However, this benefit decayed faster (at interval >2 s) among dyslexics (red) than among controls (blue; p<0.005). Error bars denote standard error (SEM).

more intervening words). Overall, dyslexics' RTs were longer than controls' (controls — $n = 29$; mean RT per word ± SEM = 725 ± 28 ms; dyslexics — $n = 23$; mean RT ± SEM = 929 ± 47 ms; $z = 3.3$, p<0.001. Mann-Whitney U test) and they were less accurate (controls — mean % correct ± SEM = 98.6 ± 0.3; dyslexics — 96.0 ± 1.4; $z = 3.2$, p<0.005. Mann-Whitney U test), as expected.

Both groups showed large recent context (repetition) effects. Specifically, both groups showed substantially faster reading RTs in the second than in the first encounter with the same non-word, when the words were proximal in time (<2 s interval), even though they were never consecutive. The smallest interval between two presentations of the same non-word was with one intervening word. As word presentation rate was based on reading rate, and dyslexics are slower readers, we equated the inter-word time interval in the two groups by considering only trials in which the second presentation occurred within <2 s ITI (offset of the first presentation to the onset of the second presentation of the same non-word). In these trials, dyslexics' benefit from repetition was similar to controls' (*Figure 5B*; mean benefit ± SEM for controls — 25% ± 2% or 170 ± 19 ms, for dyslexics — 23% ± 3% or 177 ± 20 ms; $z = 0.1$, *n.s.* Mann-Whitney U test). However, in trials with larger inter-word intervals, dyslexics' benefits were smaller. Importantly, dyslexics' benefits were smaller than controls', even though their starting point had longer RTs, as explained above (mean ITI of 42 s, max. ITI of 163 s; mean benefit ± SEM for controls — 12% ± 1% or 90 ± 10 ms, for dyslexics — 6% ± 1% or 52 ± 8 ms, $z = 3$, p<0.005. Mann-Whitney U test). Indeed, among dyslexics, the difference in benefit from repetitions with small (<2 s) compared to larger (>2 s) intervals was greater than this difference among controls ($z = 2.2$, p<0.05; Mann-Whitney U test).

Note that for short inter-word intervals, dyslexics' RT benefits do not differ from controls' in spite of an intervening word. Thus, here too, dyslexics' faster decay is consistent with a time-based effect rather than with an enhanced interference effect.

## Discussion

We characterized the dynamics of context effects, and found that they differ consistently between dyslexics and good readers. Dyslexics' behavioral benefits declined faster, both in the context of reading and in sound discrimination. In both groups, the behavioral dynamics reflected the dynamics of cortical adaptation, which decayed faster in dyslexics, both when actively performing a task and in passive listening.

### Accounting for dyslexics' 'slower processing speed'

We propose that dyslexics' shorter time constant of adaptation reflects a shorter time constant of implicit sound integration. Dyslexics' faster decay decreases the time constant of their integration of sound statistics, and consequently reduces the reliability of their implicitly calculated priors. Owing to their noisier priors, attaining the same level of processing reliability requires the extraction of more on-line information, which requires more processing time, leading to the stereotypic 'slower processing time' in dyslexia.

This account explains dyslexics' often poorer performance on rapidly presented brief auditory stimuli for both simple tones (*Tallal, 1980*; *Temple et al., 2000*) and speech (*McArthur and Bishop, 2005*; *Boets et al., 2007*). Moreover, it resolves seemingly inconsistent observations in dyslexics' pattern of perceptual difficulties. On the one hand, dyslexics' poor short term memory skills are amply documented (*Beneventi et al., 2010*; *Banai and Yifat, 2012*) and suggest that their performance should improve when memory load is reduced by decreasing task's retention intervals. On the other hand, dyslexics are known to perform poorly with short stimuli presented at brief intervals (*Tallal, 1980*), suggesting that they should gain from longer inter-stimulus intervals.

We claim that these seemingly contradictory observations result from the same basic impairment. The key feature is the proportion of on-line computations that need be allocated to solve the task. The more predictable the stimuli are for controls, both locally within session and globally in terms of long-term regularities (such as in language) the greater relative slowness ('slow processing speed') dyslexics are expected to show (consistent with *Ahissar et al., 2006*). This slowness stems from their poor implicit memory mechanisms; namely, their faster decay of the neural trace that integrates the statistics of recent stimuli.

## Dyslexics' deficit as a process rather than a representational impairment

Dyslexics' phonological and reading skills reflect the outcome of long-term learning processes which are difficult to track. The use of a simple task of discrimination of pure tones (which are physically simple but very infrequent environmentally) enabled us to observe the dynamics of context effects in a relatively novel situation. It served to dissociate the benefits of very short-term (<2 s; the scale attributed to working memory processes; *Baddeley, 2010*) implicit effects from somewhat longer-term effects. Dyslexics only benefitted less from prior statistics in the latter case.

Dyslexics' noisier priors (predictions based on past experiences) may lead to slower long-term learning of characteristic sound regularities, such as those in the native language (*Nicolson et al., 2010*). It follows that dyslexics should benefit less than controls from a given number of experiences, and that group differences should increase with additional specific experiences. Thus, although both good and poor readers are expected to improve with practice, dyslexics' benefits per exposure are expected to be smaller. Hence, counterintuitively, dyslexics' relative difficulties are expected to be greater for highly trained stimuli.

Our observation of reduced repetition effects in dyslexics' reading rate is compatible with imaging observations by *Pugh et al. (2008)*, who compared the impact of word repetition in reading on BOLD activation in posterior reading-related areas, in dyslexics versus controls. They found that although both populations gained from such repetitions, the impact differed between groups. Whereas controls' activation was monotonically reduced with repetitions, dyslexics' activation initially increased. They interpreted this pattern in a manner consistent with ours by suggesting that dyslexics' pattern seemed similar to that expected after a lesser amount of exposure to these words; i. e., slower long-term learning of these (sound) patterns. Interestingly, dyslexics' slower learning rate per event (or exposure) was also proposed by *Nicolson et al. (2010)*, who based their argument on slower learning curves in a different context. They found that dyslexics exhibit slower learning of simple motor tasks, which perhaps tap mechanisms that are partially common to those used in serial comparison tasks.

## The complex relationships between mechanisms and sites

We found that dyslexics' P2 component had a shorter adaptation duration under both active and passive conditions. We also found that their N1 had a shorter adaptation duration under the active condition when compared to controls. The P2 (which peaks 200 ms after tone onset) component was shown to reflect the accumulation of sound statistics (*Tremblay et al., 2010*; *Jaffe-Dax et al., 2015*) automatically, without explicit attention (*Sheehan et al., 2005*). Moreover, it was shown to be abnormal among dyslexics (*Bishop and McArthur, 2004*; *Jaffe-Dax et al., 2015*). Indeed, its dynamics matched those of the behavioral context effects measured simultaneously. However, even the earlier N1 component (100 ms) had a shorter time constant here among dyslexics under the active condition. Whether these two components originate from one or two cortical generators is still unclear (e.g. *Mayhew et al., 2010*; *Lanting et al., 2013*). Nevertheless, recent studies have found that the magnitudes of adaptation of both are sensitive to the statistics that characterize the experiment (e.g. *Herrmann et al., 2015*). We found a shorter time constant of adaptation in N1 than in P2.

Our observations suggest that the dynamics of adaptation in the auditory cortex of dyslexics differs from that of controls. But they do not directly point to the anatomical source of this abnormality as 100–200 ms ERP signals reflect the combined contribution of various brain sites. Thus, although dyslexics' processing deficit is likely to result from a structural variation (which probably has a genetic origin; *Giraud and Ramus, 2013*), our observations are consistent with several accounts. The abnormally fast decay of N1 and P2 adaptation may reflect a different anatomy of the auditory cortex, as suggested in a recent longitudinal study (*Clark et al., 2014*), or impaired long-range connectivity between posterior and frontal areas (*Boets et al., 2013*; *Ramus, 2014*), which are not mutually exclusive.

With respect to nature of dyslexics' auditory processing deficits, several accounts are consistent with our current findings. These include both impaired top-down control (*Díaz et al., 2012*), and unreliable (i.e. more variable) auditory responses (*Hornickel and Kraus, 2013*) associated with seemingly noisier sensory systems (*Sperling et al., 2005*), particularly in some frequency bands (*Goswami et al., 2002*; *Goswami, 2011*). All these observations are expected outcomes of poorer

implicit predictions made on the basis of stimuli statistics. Indeed, when dyslexics' behavioral sensitivity to complex and challenging noise stimuli was assessed, it was found that their impairment is specific for repeated stimuli (with >2 sec intervals). Moreover, their performance tended to be even better than controls' when stimuli differed from the repeated, and hence implicitly predicted, ones (*Daikhin et al., 2016*; for visual analog see also *Jaffe-Dax et al., 2016*).

Importantly, the involvement of the auditory cortex does not exclude the contribution of sub-cortical regions. Both the cerebellum and the basal ganglia are involved in the process of integrating sound regularities into improved perceptual performance with reduced reliance on on-line working memory processes and increased reliance on specific sound predictions (*Daikhin and Ahissar, 2015*). Thus, the hypothesis linking dyslexics' reduced rate of automatization (in reading and in other tasks) with impaired usage of cerebellar processes (*Nicolson and Fawcett, 1990*; *Nicolson et al., 2010*) is also in line with our findings and interpretation.

The attempts to map dyslexics' processing difficulties to several potentially relevant anatomical abnormalities suggest that there may not be a single anatomical source of difficulty, and that the issue of the core deficit underlying dyslexia should be seen as process-related rather than structure-related. Our observation that the process of dyslexics' neural adaptation is abnormally short paves the way for future studies of the neural processes that underlie reading difficulties in non-humans. An important insight suggested by our study, which is further supported by recent human studies in speech perception (e.g. *Kleinschmidt and Jaeger, 2016*), is that adaptation reflects a mechanism for statistical learning.

This interpretation is in line with recent animal studies, suggesting that adaptation reflects a mechanism for making implicit high-resolution stimulus predictions on the basis of an experiment's statistics (*Khouri and Nelken, 2015*). Another studied aspect of neural adaptation is its occurrence at multiple time scales, as observed both in animals (*Ulanovsky et al., 2004*; *Khouri and Nelken, 2015*) and in humans (reviewed in *Lu and Sperling, 2003*). Here we propose that dyslexics' deficit is characterized at time scales >2 s after stimulus presentation. It would be interesting to compare whether individuals with memory deficits at shorter time intervals (e.g. in iconic/anechoic memory, i.e.<500 ms) have broader cognitive deficits, as suggested by previous studies (*Lu et al., 2005*; *Miller et al., 2010*).

## The prevalence of shorter adaptation in dyslexia

About half of our dyslexic participants showed adaptation time scales that were within the range of controls (*Figures 2–4*). This partial overlap between the two groups could have stemmed from the low reliability of our measurements. Another potential source for this variability might be different types of dyslexia among our dyslexic participants, which may relate to different underlying neural mechanisms. The standard reading tests that were used for inclusion or exclusion of participants might not have been fine-grained enough to define participants along the dimensions that map to different sources of underlying difficulties (*Zoccolotti and Friedmann, 2010*).

## The putative functional relationship between shorter adaptation and long-term effects

We only tracked the short-term effects of dyslexics' faster decay of memory trace. Therefore, we can only speculate about the relationship between these observations and dyslexics' difficulties in acquiring expert-level proficiency in reading. One putative conceptual link may be provided by the Bayesian framework, when the principle of efficient coding is introduced (*Wei et al., 2015*). This principle predicts that more likely stimuli gradually acquire denser, more reliable representations. The typical gradual enhancement of the representation of more likely stimuli may be reduced in dyslexia, leading to reduced sensitivity to the specific phonological and morphological forms that characterize native language. Assessing this hypothesis experimentally requires long-term tracking of the learning of novel sound statistics.

# Materials and methods

## Participants

Sixty native Hebrew speakers (30 dyslexics and 30 good readers), all of whom were students at the Hebrew University [mean age (STD) = 24.2 (5.4) years; 36 females] were recruited for this study. Recruitment was based on ads at the Hebrew University. Monetary compensation for participation was according to standard student rates. The study was approved by the Hebrew University Committee for the Use of Human Subjects in Research. All dyslexic participants had been diagnosed as having a specific reading disability by authorized clinicians. Reading-related measures were also assessed in our lab (detailed in *Table 1*). Participants with more than 2 years of formal musical education were excluded, so that musical training would not be a major contributor to their pitch sensitivity (*Micheyl et al., 2006*; *Parbery-Clark et al., 2011*). Participants with poor Block Design scores (lower than a normalized score of 7) were also excluded from the study. All participants filled in a questionnaire regarding any neurological or psychiatric disorders. None of the participants reported any such disorders. None of them had ever participated in a similar auditory experiment in our lab.

## Experimental procedure

Participants were administered four sessions on four different days.

In session one, participants were administered a series of cognitive assessments. Thirty dyslexics and 30 controls were admitted to this session.

In session two, participants performed a two-tone frequency discrimination task with a specially designed sequence of trials (Experiment 1; *Figure 1A–B*). The same 30 dyslexics and 30 controls participated in this session.

In session three, ERPs were recorded both passively and while performing the discrimination task. First, participants watched a silent movie while a series of single tones was presented to them in four blocks of four different Inter Stimulus Intervals (ISI) of 2, 3.5, 6.5 or 9.5 s (Experiment 2b), in a random order. Second, participants actively engaged in the two-tone frequency discrimination task in four blocks with different Inter-Trial-Intervals (ITI – time interval between the second tone in the trial and first tone in the following trial) of 1.4, 2.9, 5.9 and 8.9 s (Experiment 2a). The ISI between

**Table 1.** Cognitive scores for the dyslexic and control groups (mean and standard deviation). Dyslexics differed from their good-reader peers in all phonological tasks and in verbal working memory, but not in their general reasoning skills (Mann-Whitney U tests).

| Test | Control (STD) N = 30 | Dyslexic (STD) N = 30 | Mann-Whitney *z* value |
|---|---|---|---|
| Age (years) | 25.8 (3.0) | 24.3 (3.1) | 1.6 |
| General cognitive (scaled) | | | |
| Block Design | 12.4 (2.9) | 12.1 (3.5) | 0.3 |
| Digit Span | 11.1 (2.8) | 7.8 (1.7) | 4.7*** |
| Phonological speed [items/minute] | | | |
| Pseudo-word reading rate | 58.4 (24.4) | 32.2 (10.5) | 4.4*** |
| Single-word reading rate | 96.8 (32.5) | 68.3 (25.8) | 3.3** |
| Word pattern recognition rate | 68.1 (15.2) | 39.9 (13.5) | 5.6*** |
| Passage reading rate | 140.4 (23.8) | 97.8 (22.3) | 5.7*** |
| Spoonerism rate | 10.0 (3.0) | 5.8 (3.2) | 4.6*** |
| Phonological accuracy [% correct] | | | |
| Pseudo-word reading accuracy | 89.7 (11. 2) | 62.4 (18.3) | 5.1*** |
| Single-word reading accuracy | 97.1 (4.3) | 87.6 (8.3) | 4.8*** |
| Word pattern recognition accuracy | 100 (0) | 96.2 (6.4) | 4.3*** |
| Passage reading accuracy | 98.6 (1.4) | 94.8 (4.5) | 4.9*** |
| Spoonerism accuracy | 92.2 (6.8) | 77.9 (18) | 3.2** |

*p<0.05; **p<0.005; ***p<0.0005.

the two tones in the trial was 600 ms. Thus, in both passive and active conditions, the Stimulus Onset Asynchronies, i.e. the time intervals between the onset of the first tones in adjacent trials were 2, 3.5, 6.5 and 9.5 s. A subgroup of 25 dyslexics and 23 controls participated in this session.

In session four, participants performed a fast reading task of visually presented single non-words. Voice response was recorded (Experiment 3; *Figure 4A*). Both rate and accuracy were obtained. 29 dyslexics and 23 controls participated in this session.

All sessions were administered in a sound-attenuated room. Sounds and visual presentations were produced using Matlab (The Mathworks, Inc., Natick, MA). Tones were presented and voice response was recorded by Psychtoolbox and Psychportaudio (*Kleiner et al., 2007*) through a Saffire 6 USB audio interface (Focusrite Audio Engineering ltd., High Wycombe, UK).

## Cognitive assessments

General cognitive abilities and phonological skills were assessed using standard tasks:

A. Non-verbal reasoning ability. This was measured with the Block Design, a standard test for assessing visuo-spatial reasoning (WAIS-III; *Wechsler, 1997* ).

B. Short-term verbal memory. This was evaluated with the standard Digit Span task (forward and backward; Hebrew version of WAIS-III; *Wechsler, 1997*).

C. Phonological decoding and single-word reading. Pseudo-word and single word reading were assessed using standard Hebrew lists designed by *Deutsch and Bentin (1996)*.

D. Word pattern recognition. Subjects were presented with 24 pairs, each composed of a word and a pseudo-homophone, and were asked to point to the word in each pair.

E. Fluent reading. Subjects read an academic level passage of 150 words followed by a comprehension question.

F. Phonological awareness was assessed using the Spoonerism task (*MacKay, 1970*; *Möller et al., 2007*). Participants heard (Hebrew) word pairs and were asked to switch the first phonemes of the two words and respond vocally (e.g.: /laila tov/, 'good night' in Hebrew, should be switched to /taila lov/).

In all phonological and reading tasks, both accuracy and rate were scored.

## Stimuli for dissociating between context effects of recent versus earlier trials – Experiment 1

Participants performed four blocks of 150 trials of the two-tone frequency discrimination task. Each trial contained a tone pair (50 ms, 70 dB each tone; 600 ms inter-tone intervals). They were asked to indicate which of the two tones had a higher pitch. A short demo of 10 trials preceded the actual experiment. Feedback was provided only in the demo trials. 80% success on the 10 demo (easy) trials was a prerequisite for continuing the task. We did not administer feedback during the assessment because we did not want to affect the magnitude of the listeners' contraction bias.

The task was administered with a set of constant stimuli that we designed specifically for this experiment (available at: https://goo.gl/UnrG1A). Its design allowed us to evaluate the contribution to the context effect of the most recent trial separately from that of all previous trials. Assessing these effects separately required a specifically designed sequence, since these effects are typically correlated. Specifically, the direction of the frequency distance between the first tone of the current trial and that of the first tone of the most recent trial, and the direction of the distance from that tone to the average across trials, are typically correlated. In the design of this series, we ensured that they were not correlated. In other words, the sign of the global context ($G$): $G(t) = sign(f_1(t) - \langle f_1 \rangle)$ and that of the recent context ($R$): $R(t) = sign(f_1(t) - f_1(t-1))$, in each trial were not correlated. In this sequence, the overall contribution of both the local and global context was positive, yet small.

## Stimuli for assessing the impact of time intervals — Experiments 2a and 2b

In the active condition (Experiment 2a), participants were administered four blocks of 100 trials each of the two-tone frequency discrimination task. Each trial contained a tone pair (50 ms, 70 dB each tone; 600 ms inter-tone intervals), and listeners were asked to indicate which of the two tones had a higher pitch. The sequence of trials was randomly drawn for each participant. In each trial, a tone

was chosen randomly from 800 Hz to 1250 Hz. The other tone was chosen randomly to have a frequency difference (plus or minus) between 1% and 30% from the previously chosen tone. The order of the tones was also randomly chosen. Trial onset asynchrony was fixed for each block at 2, 3.5, 6.5 or 9.5 s. Block order was counterbalanced across subjects.

In the first, passive, part of the session (Experiment 2b), only the first tone in each pair was presented. We compensated for it by increasing the Inter-Stimulus Intervals (between the tone's offset on the previous trial and the onset of the current trial) in this condition by 0.6 s. Consequently, the onset-to-onset intervals between first tones of adjacent events were the same in the two conditions. Subjects watched a silent movie and were asked to ignore the tones.

## Stimuli for assessing the dynamics of benefits in oral reading rate — Experiment 3

Participants were presented with six blocks of 120 non-words and were asked to read them aloud as fast as they could. Voice onset and offset were acquired. Each non-word was presented 500 ms after the voice offset of the preceding non-word. Presentation remained until the voice offset of the current non-word. Non-words were randomly generated by conjunction of two randomly chosen valid Hebrew syllables (consonant-vowel and consonant-vowel-consonant, or vice versa).

## ERP recordings and analyses

Electrophysiological activity was recorded from 32 active Ag-AgCl electrodes mounted on an elastic cap using the BioSemi ActiveTwo tools and recording software (BioSemi B.V., Amsterdam, The Netherlands). Electrode sites were based on the 10–20 system (American Electroencephalographic Society, 1991). Two additional electrodes were placed over the left and right mastoids. Horizontal EOG was recorded from two electrodes placed at the outer canthi of both eyes. Vertical EOG was recorded from electrodes on the infraorbital and supraorbital regions of the right eye in line with the pupil.

EEG and EOG signals were amplified, filtered with an analogue band-pass filter of 0.16–100 Hz, and sampled at 256 Hz. Offline analysis was performed using Brain Vision Analyzer 1.05 software (Brain Products GmbH, Gilching, Germany) and EEGLAB toolbox for Matlab (*Delorme and Makeig, 2004*). The EEG signal was digitally band-pass filtered between 1 Hz and 30 Hz to remove large drifts in signal and high-frequency noise. ICA analysis was trained on the entire length of each block and on all scalp electrodes to identify components that reflect eye-blink- or eye-movement-evoked electrical activity. An eye-related component was identified by its time-correlation with the occurrence of blinks or saccades. This relationship between the identified component and eye-blink activity was verified by confirming that the component's scalp distribution was typical of eye-related electrical activity (*Delorme et al., 2007*). Data were referenced to the nose channel to remove external electrical influence. Artifact rejection was applied to the non-segmented data according to the following criteria: any data point with an EOG or EEG > ± 100 μV was rejected along with the data ± 300 ms around it. In addition, if the difference between the maximum and the minimum amplitudes of two data points within an interval of 50 ms exceeded 100 μV, data ± 200 ms around it were rejected. Finally, if the difference between two adjacent data points was more than 50 μV, the data ± 300 ms around it were rejected. Trials containing rejected data points were omitted from further analysis. Groups did not differ on the number of trials that were analyzed (active condition: controls — 387 ± 6; dyslexics — 382 ± 5; $z = 1.6$, *n.s.*; passive condition: controls — 394 ± 4; dyslexics — 394 ± 4; $z = -0.2$, *n.s.*; mean number of trials ± SEM, Mann-Whitney U tests).

For ERP averaging across trials, the EEG was parsed to 2000 ms epochs starting 500 ms before the onset of the first stimulus in each pair, and averaged separately for each electrode. The baseline was adjusted by subtracting the mean amplitude of the pre-stimulus period (500–150 ms before the onset of the first stimulus in the trial) of each ERP from each data point in the epoch. The pre-stimulus baseline period was calculated from this time interval to exclude effects of anticipatory responses that preceded informative anticipated stimuli (CNV; *Walter et al., 1964*).

ERP analysis was based on the epochs that were recorded with electrode Cz (at the vertex of the scalp). This electrode measured the most prominent response to the auditory stimuli. Data from each acquisition session were analyzed separately. The magnitude of the ERP components was calculated as the area under the curve between 70 ms and 130 ms after first tone's onset for N1 and

150–250 ms time range for P2. We repeated the entire analysis for adjacent electrodes (Fz, FC1, FC2, C3, C4, CP1, CP2 and Pz) and found similar results.

## Estimation of the dynamics of recovery from adaptation

Context effects and ERPs in each of the four different ITIs (experiment 2) were fitted with an exponential decay model for each participant separately. This model was previously used to characterize the decay of context effects both behaviorally and for MEG measurements (*Lu et al., 1992*; *Sams et al., 1993*). In a previous study, the effect of context as a function of number of trials back was quantitatively measured and indeed resembled an exponential decay (*Raviv et al., 2012*). Following that quantitative description, we found that the exponential decay of previous trials captures context effects during this task (*Jaffe-Dax et al., 2015*). In the current study, we modeled the impact of previous trials as a function of the temporal interval instead of number of trials. The model, $\alpha + \beta exp(-t/\tau)$, had three parameters: $\alpha$ – asymptote after recovery; i.e., the value expected when $t \rightarrow \infty$, $\beta$ – the magnitude of adaptation; i.e., the value expected at $t = 0$ minus $\alpha$, and $\tau$ – time constant of adaptation; i.e., the time it takes for the measure expected at $t = 0$ to decay to $1/e$ (~37%) of its initial value. A small $\tau$ indicates fast decay and a large $\tau$ indicates a slow decay.

Formally, we searched for the triplet of parameters that minimizes the squared difference between the data and the model prediction: the units of $\alpha$ and $\beta$ are those of the fitted measure ($d'$ for behavioral bias, $\mu V \, x \, msec$ for ERP). In the passive condition, $t$ is the Inter-Stimulus Interval (ISI) between the offset of a tone and the onset of the consecutive tone. In the active condition, $t$ is the Inter-Trial Interval (ITI) between the offset of the second tone and the onset of the first tone in the consecutive trial. Fitted parameters were estimated by minimizing the squared error of the exponential decay model in a limited range. For the ERP adaptation, limits were from 0 to 15,000 for $\alpha$ and from $-15,000$ to 0 for $\beta$. For the $d'$ difference, the limits were from 0 to 100 for $\alpha$ and $\beta$. For both measures, $\tau$ was limited to be from 0 to 100 s. The groups did not differ on the exponential curve's Goodness-of-Fit for any of the analyzed measurements ($z < 1.8$, n.s. Mann-Whitney U tests). The exponential decay model captured the ERP adaptation decay well for both groups ($R^2 > 0.4$). The model was less able to account for the behavioral bias decay, especially for controls ($R^2 = 0.2$), suggesting a more complex mechanism than could be well described by a single exponential decay.

For purposes of compatibility with our previous studies (*Raviv et al., 2012*; *Jaffe-Dax et al., 2015*) and to avoid assumptions of normal distribution, we used conservative, non-parametric tests throughout the study. Using standard parametric tests yielded similar statistical significance.

## Acknowledgements

We thank Ofri Raviv for his useful advice in the creation of the stimuli for Experiment one and Elad Goldfarb for his help in verifying word onset times in Experiment 3. This study was supported by the Israel Science Foundation (ISF grant no. 616/11 and Canada-Israel grant no. 2425/15), the Gatsby Charitable Foundation, The German-Israeli Foundation for Scientific Research and Development (grant no. I-1303–105.4/2015), Canadian Institutes of Health Research (CIHR), The International Development Research Center (IDRC) and the Azrieli Foundation.

## Additional information

### Funding

| Funder | Grant reference number | Author |
|---|---|---|
| Israel Science Foundation | 616/11 | Merav Ahissar |
| Gatsby Charitable Foundation | | Merav Ahissar |
| German-Israeli Foundation for Scientific Research and Development | I-1303-105.4/2015 | Merav Ahissar |
| Canadian Institutes of Health Research | | Merav Ahissar |
| International Development Re- | | Merav Ahissar |

search Centre

| Azrieli Foundation | | Merav Ahissar |
|---|---|---|
| Israel Science Foundation | 2425/15 | Merav Ahissar |

The funders had no role in study design, data collection and interpretation, or the decision to submit the work for publication.

## Author contributions

SJ-D, Conceptualization, Software, Formal analysis, Validation, Investigation, Visualization, Methodology, Writing—original draft, Writing—review and editing; OF, Investigation, Writing—review and editing; MA, Conceptualization, Supervision, Funding acquisition, Validation, Investigation, Methodology, Writing—original draft, Writing—review and editing

## Author ORCIDs

Sagi Jaffe-Dax, http://orcid.org/0000-0002-8759-6980

## Ethics

Human subjects: Informed consent was acquired from all participants. The study was approved by The Hebrew University Committee for the Use of Human Subject in Research.

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
