## [Decision Letter]

Thank you for submitting your article "Shorter neural adaptation to sounds accounts for dyslexics' abnormal perceptual and reading dynamics" for consideration by *eLife*. Your article has been reviewed by three peer reviewers, and the evaluation has been overseen by Andrew King as the Reviewing and Senior Editor. The following individuals involved in review of your submission have agreed to reveal their identity: John Stein (Reviewer #1); Zhong-Lin Lu (Reviewer #2); Anne-Lise Giraud (Reviewer #3).

The reviewers have discussed the reviews with one another and the Reviewing Editor has drafted this decision to help you prepare a revised submission.

Summary:

The authors have carried out a novel and coherent series of studies that provide clear evidence that dyslexic individuals show a weaker effect of prior stimuli in a simple two tone discrimination task. By manipulating the temporal interval between tones in a sequence while carrying out behavioral measurements and analyses of event related potentials, they demonstrated that the dyslexic participants showed less of the 'anchoring effect' attributable to the previous trials, implying that implicit memory decays faster relative to controls. A third study demonstrated a diminished benefit of re-presentation of nonword stimuli for the dyslexic participants. The authors describe the effects as 'shorter neural adaptation' and hypothesize that they reflect a 'noisier' neural environment, which can result in the cumulative difficulties that lead to reading disability.

Essential revisions:

The reviewers agree that this is a well designed and conducted study and that the behavioral and ERP data provide clear evidence during active discrimination, passive listening and reading for faster decay of implicit memory in dyslexics than in controls. This is potentially an extremely interesting result. They have, however, raised several issues that will need to be addressed in a revised version of the manuscript.

1) The first major point that could be clarified concerns the assessment of the dynamics of P2 and N1 in relation to the behavioral curve. Based on a previous study, the authors assume that only P2 is directly associated with the contraction bias (i.e. the perceptual contraction of the pitch of the target toward the mean pitch of previous stimuli). However, dynamics of N1 seems to better fit the behavioral curve than P2. The decrease in P2 and the decrease in contraction bias are fitted with an exponential decay model, from which three parameters, the 'temporal slope parameter' (representing the decay), the decay magnitude and the asymptotic level are estimated to minimize the squared difference between the data and the prediction model. The three parameters estimated for N1 dynamics (Figure 3) appear very similar to the behavioral bias dynamics (Figure 2 bottom). They seem more dissimilar for P2. This point needs to be clarified. Further comment is also needed about the lack of difference in the time constant of adaptation for N1 between dyslexics and controls in the passive listening condition, given that adaptation appears to be interpreted here as an automatic process. It might be better if Figure 2 and Figure 3 are shown with the same scale.

2) The second major point relates to the relation between altered implicit memory and cognitive and phonological skills in dyslexia. As mentioned by the authors in “The prevalence of shorter adaptation in dyslexia”, about half of the dyslexic participants showed adaptation time scales that were within the range of controls. First, perhaps these individual data should appear as supplementary data. Second, is there any correlation between adaptation time scales (both in the ERP data and behaviorally) and cognitive abilities/phonological skills, such as short-term memory, phonological decoding/awareness and fluent reading? This analysis would add usefully to the data from the reading experiment and may shed light on the differences observed for presentations intervals of less than or greater than 2 seconds.

3) More discussion of the possible physiological basis of this faster decay is needed. In particular, earlier MEG studies have shown that the dependence of event-related fields on inter-stimulus interval is due to habituation (Lu & Sperling, 2003, Measuring sensory memory: MEG habituation and psychophysics, In Magnetic Source Imaging of the Human Brain, Edited by Z-L Lu and L. Kaufman. Lawrence Erlbaum Associates, Inc., Mahwah, New Jersey. Page 319-342.), engaging physiological mechanisms that are different from adaptation. The authors should address the possibility that habituation may provide a better explanation for their data.

4) The results should be considered in the context of general cognitive deficits, since previous work has shown that faster decay of sensory memory is related to early signs of Alzheimer's disease (Lu, et al., 2005, Fast decay of iconic memory in observers with mild cognitive impairments. PNAS, 102, 1797-1802) and lower psychometric intelligence (Miller et al., 2010, Decay of iconic memory traces is related to psychometric intelligence: A fixed-links modeling approach. Learning and Individual Differences, 20, 699-704).

---

## [Author Response]

*Essential revisions:*

*The reviewers agree that this is a well designed and conducted study and that the behavioral and ERP data provide clear evidence during active discrimination, passive listening and reading for faster decay of implicit memory in dyslexics than in controls. This is potentially an extremely interesting result. They have, however, raised several issues that will need to be addressed in a revised version of the manuscript.*

*1) The first major point that could be clarified concerns the assessment of the dynamics of P2 and N1 in relation to the behavioral curve. Based on a previous study, the authors assume that only P2 is directly associated with the contraction bias (i.e. the perceptual contraction of the pitch of the target toward the mean pitch of previous stimuli). However, dynamics of N1 seems to better fit the behavioral curve than P2. The decrease in P2 and the decrease in contraction bias are fitted with an exponential decay model, from which three parameters, the 'temporal slope parameter' (representing the decay), the decay magnitude and the asymptotic level are estimated to minimize the squared difference between the data and the prediction model. The three parameters estimated for N1 dynamics (Figure 3) appear very similar to the behavioral bias dynamics (Figure 2 bottom). They seem more dissimilar for P2. This point needs to be clarified. Further comment is also needed about the lack of difference in the time constant of adaptation for N1 between dyslexics and controls in the passive listening condition, given that adaptation appears to be interpreted here as an automatic process. It might be better if Figure 2 and Figure 3 are shown with the same scale.*

Indeed, based on our previous study (Jaffe-Dax, et al., 2015, J Neurosci), we hypothesized that the difference between the groups would be found in the dynamics of P2 adaptation, as indeed was the case for both the passive and active conditions. Given other people findings (Hermann, et al., 2015, J Neurophysiol), that N1 is also sensitive to experiment’s statistics we further assessed dyslexics N1 dynamics and found that in the active, but not in the passive condition, this component also has faster decay compared with controls. This pattern of observations is consistent with our hypothesis. However, we now address the complexity (in terms of interpretation) induced by the additional, somewhat mixed findings with N1. These points are now addressed in subsections “Assessing the dynamics of the neural trace” and “Comparing the dynamics of passive adaptation between controls and dyslexics”.

Re scales in figures – thanks: Figure 2, Figure 3 and Figure 4 (previously notated as 2B, 3B and 4B respectively) are now shown with the same scale.

*2) The second major point relates to the relation between altered implicit memory and cognitive and phonological skills in dyslexia. As mentioned by the authors in “The prevalence of shorter adaptation in dyslexia”, about half of the dyslexic participants showed adaptation time scales that were within the range of controls. First, perhaps these individual data should appear as supplementary data. Second, is there any correlation between adaptation time scales (both in the ERP data and behaviorally) and cognitive abilities/phonological skills, such as short-term memory, phonological decoding/awareness and fluent reading? This analysis would add usefully to the data from the reading experiment and may shed light on the differences observed for presentations intervals of less than or greater than 2 seconds.*

Regarding single participants’ data – these are now uploaded to our website: https://goo.gl/UnrG1A. We shall be happy to send them as supplementary data, if you think it is preferable.

Regarding correlations – We indeed found a significant correlation, among controls, between the rate of decay of their contraction bias (behavioral τ) and their reading benefit at the longer time interval (i.e. faster decay correlated with smaller reading benefits at long temporal intervals; Spearman’s *R* = 0.48, p < 0.05), though not among dyslexics (Spearman’s *R* = -0.28, n.s.). But, the significance of this correlation is not sufficient (is not resilient to correcting for additional post-hoc comparisons), given that we also assessed correlation with time constants of N1 and P2 adaptation, which were not significant. Choosing a conservative approach, we thus decided not to address this correlation (plotted in the figure below) in our manuscript.

Author response image 1.Cross-subject correlation between behavioral time constants of implicit memory decay and behavioral benefits in reading a repeated non-word, when repetition occurred with more than 2 seconds intervals.Left: Significant correlation among controls (Spearman’s *R* = 0.48, *p* < 0.05). Right: no correlation among dyslexics (Spearman’s *R* = -0.28, *n.s.*). Data shown as rank order to correspond with the statistical test which was applied.**DOI:**
http://dx.doi.org/10.7554/eLife.20557.009

*3) More discussion of the possible physiological basis of this faster decay is needed. In particular, earlier MEG studies have shown that the dependence of event-related fields on inter-stimulus interval is due to habituation (Lu & Sperling, 2003, Measuring sensory memory: MEG habituation and psychophysics, In Magnetic Source Imaging of the Human Brain, Edited by Z-L Lu and L. Kaufman. Lawrence Erlbaum Associates, Inc., Mahwah, New Jersey. Page 319-342.), engaging physiological mechanisms that are different from adaptation. The authors should address the possibility that habituation may provide a better explanation for their data.*

We now further elaborate on the potential neural mechanisms probed by the decay of N1 and P2 and their relation to prediction formation in subsection “The complex relationships between mechanisms and sites”.

*4) The results should be considered in the context of general cognitive deficits, since previous work has shown that faster decay of sensory memory is related to early signs of Alzheimer's disease (Lu, et al., 2005, Fast decay of iconic memory in observers with mild cognitive impairments. PNAS, 102, 1797-1802) and lower psychometric intelligence (Miller et al., 2010, Decay of iconic memory traces is related to psychometric intelligence: A fixed-links modeling approach. Learning and Individual Differences, 20, 699-704).*

These studies address very short time constants of memory decay (e.g. iconic memory decays with few hundred ms). We now discuss the concept of hierarchy of time constants of implicit memory. We propose that deficits in shorter time constants may be more detrimental to cognitive skills; subsection “The complex relationships between mechanisms and sites”.